# Trends and patterns of antibiotic prescribing at orthopedic inpatient departments of two private-sector hospitals in Central India: A 10-year observational study

**Kristina Skender**[ID][1], **Vivek Singh**[2], **Cecilia Stalsby-Lundborg**[ID][1], **Megha Sharma**[ID][1,3]*

**1** Department of Global Public Health, Health Systems and Policy, Karolinska Institutet, Stockholm, Sweden, **2** Department of Orthopedic, Ruxmaniben Deepchand Medical College, Ujjain, India, **3** Department of Pharmacology, Ruxmaniben Deepchand Gardi Medical College, Ujjain, India

* meghasharma27@rediffmail.com

## Abstract

### Background

Frequent antibiotic prescribing in departments with high infection risk like orthopedics prominently contributes to the global increase of antibiotic resistance. However, few studies present antibiotic prescribing patterns and trends among orthopedic inpatients.

### Aim

To compare and present the patterns and trends of antibiotic prescription over 10 years for orthopedic inpatients in a teaching (TH) and a non-teaching hospital (NTH) in Central India.

### Methods

Data from orthopedic inpatients (TH-6446; NTH-4397) were collected using a prospective cross-sectional study design. Patterns were compared based on the indications and corresponding antibiotic treatments, mean Defined Daily Doses (DDD)/1000 patient-days, adherence to the National List of Essential Medicines India (NLEMI) and the World Health Organization Model List of Essential Medicines (WHOMLEM). Antibiotic prescriptions were analyzed separately for the operated and the non-operated inpatients. Linear regression was used to analyze the time trends of antibiotic prescribing; in total through DDD/1000 patient-days and by antibiotic groups.

### Results

Third generation cephalosporins were the most prescribed antibiotic class (TH-39%; NTH-65%) and fractures were the most common indications (TH-48%; NTH-48%). Majority of the operated inpatients (TH-99%; NTH-97%) were prescribed pre-operative prophylactic antibiotics. The non-operated inpatients were also prescribed antibiotics (TH-40%; NTH-75%), although few of them had infectious diagnoses (TH-8%; NTH-14%). Adherence to the NLEMI was lower (TH-31%; NTH-34%) than adherence to the WHOMLEM (TH-65%; NTH-

**Data Availability Statement:** All data generated during the study are included in the article. As per institute's policy, the metadata of any research is not shared with general public. This is because of

the probable medical, ethical and legal issues connected with patient safety, hospital and hospital staff safety. However, the data can be made available for the researchers who meet the criteria to access the confidential data via the Chairman of the Ethics Committee, R.D. Gardi Medical College, Agar Road, Ujjain, Madhya Pradesh, India, 456006 (email: iecrdgmc@yahoo.in, uctharc@bsnl.in), by giving all details of the article. The ethical approval number: 41-2/2007, 114/2010 and 311/2013 needs to be quoted along with the request.

**Funding:** The study was funded by the Swedish Research Council (Vetenskapsrådet; K2007-70X-20514-01-3, K2010–396-70X-20514-04-3, 2017–01327) and Asia Link (348–2006-6633). KS is the recipient of scholarships by Karolinska Institutet Foundation 2019-01843 and Elisabeth and Alfred Ahlqvist's Foundation 2019 from Apotekarsocieteten, Sweden. MS is the recipient of Erasmus Mundus Lot-15. The funders had no role in study design, data collection and analysis, decision to publish, or preparation of the manuscript.

**Competing interests:** The authors have declared that no competing interests exist.

**Abbreviations:** ATC, Anatomical therapeutic chemical; DDD, Defined daily dose; FDC, Fixed-dose combination; LMIC, Low and middle-income country; MRSA, Methicillin-resistant *Staphylococcus aureus*; NLEMI, National List of Essential Medicines in India; NTH, Non-teaching hospital; TH, Teaching hospital; WHO, World Health Organization; WHOCC, World Health Organization Collaborating Centre for Drug Statistics Methodology; WHOMLEM, World Health Organization Model List of Essential Medicine.

62%) in both hospitals. Mean DDD/1000 patient-days was 16 times higher in the TH (2658) compared to the NTH (162). Total antibiotic prescribing increased over 10 years (TH-$\beta$ = 3.23; NTH-$\beta$ = 1.02).

## Conclusion

Substantial number of inpatients were prescribed antibiotics without clear infectious indications. Adherence to the NLEMI and the WHOMLEM was low in both hospitals. Antibiotic use increased in both hospitals over 10 years and was higher in the TH than in the NTH. The need for developing and implementing local antibiotic prescribing guidelines is emphasized.

## Introduction

Rational use of antibiotics is essential for reducing morbidity and mortality caused by bacterial infections. However, antibiotics are often prescribed irrationally, which leads to consequences such as adverse drug events, poorer health outcomes, waste of resources, economic burden, unnecessary contamination of the environment, and development of antibiotic resistance [1–4].

Antibiotic resistance is a serious threat to public health globally and it extensively affects health and economy specifically of low- and middle-income countries (LMICs), including India [5,6]. It is estimated that by 2050, antibiotic resistance will cause 10 million deaths per year worldwide, out of which 2 million deaths are projected in India [7]. India is one of the biggest consumers of antibiotics in the world, and despite the worldwide decrease in infectious diseases, antibiotic use in India is on the rise [8]. From 2000 to 2015, antibiotic consumption in India increased by 103%, which is more than in any other country; due to remaining infectious disease burden, overall increased access to antibiotics and misuse [8].

Focused interventions could control and minimize the misuse of antibiotics; however, specific target areas of these interventions are unidentified in many countries. The World Health Organization (WHO) recommends to monitor, register and analyze local antibiotic prescribing practices with respect to the diagnoses, and compare them with other health facilities to determine the target areas for intervention [3].

The wounds of orthopedic surgeries are often deep-seated and difficult to treat; therefore, patients are at high risk of developing healthcare-associated infections with long-term recurrence risk, especially in the case of methicillin-resistant *Staphylococcus aureus* (MRSA) [9]. The prophylactic dose of antibiotic is crucial to prevent infections related to surgical cuts and implants, and consequently to decrease the morbidity, disability and mortality in orthopedic patients [10,11]. According to the available prescribing guidelines, a prophylactic dose of antibiotic is recommended prior to surgery, the pre-operative prophylaxis. The relative infection risk is estimated to reduce by 81%, when using antibiotic prophylaxis in total knee and hip replacement surgery [12]. The choice of prophylactic antibiotics, dose, timing and duration of therapy play an important role in this reduction, however, remain controversial [10,11]. Surgical antibiotic prophylaxis is meant to reduce postoperative complications, but on the other hand, it increases the risk of antibiotic resistance [10,13]. Antibiotic resistance makes routine orthopedic surgeries more challenging, which can result in physical disabilities and life-threatening infections [14].

Although approximately 80% of healthcare facilities in India are private, research studies have been predominately conducted in public sector facilities [15–17]. In the absence of basic

data, the amount of actually prescribed antibiotics could not be estimated. Even though the national prescribing guidelines are equally applicable to both public and private healthcare facilities in India, a few studies conducted in the private sector show that private facilities often fail to implement the guidelines [15–17].

It is thus, crucial to determine the patterns of prescribing antibiotics in high infection risk departments at private sector facilities. To date, only a few studies have analyzed antibiotic prescribing patterns [18,19] and no studies exist that investigate antibiotic prescribing trends over a long time at orthopedic departments in LMICs. Therefore, this study aims to analyze, compare and present antibiotic prescription patterns and trends over 10 years period, in orthopedic departments at two private sector hospitals, to identify areas for sustaining or achieving rational use of antibiotics.

## Methods

### Study setting

Data were collected for inpatients of orthopedics departments at a teaching hospital (TH) and a non-teaching hospital (NTH) in Ujjain, Madhya Pradesh, Central India. Both hospitals are private and regulated by the same charitable trust [15–17]. The TH is associated with Ruxmaniben Deepchand Gardi Medical College and located in outskirts of Ujjain city. It has a capacity of 800 beds and provides medical services and drugs free of charge to all patients. Doctors at the TH get fixed monthly salary and visits from pharmaceutical sales representatives are restricted. The NTH is situated in the centre of the city and has a capacity of 400 beds. In the NTH, the medical services are charged to the patients and the medicines are purchased out-of-pocket. Being a hospital of a charitable trust, the charges in the NTH are lower compared to other private sector hospitals. The prescribers are paid based on the number of hospital visits made and the number of patients admitted by them. Additionally, in the NTH, there is no administrative restriction on the doctors, and they can be contacted by the pharmaceutical sales representatives. Furthermore, in the NTH, the prescribed medicines could be bought from any of the pharmacies located in the area. Both hospitals have working microbiology laboratory for antibiotic susceptibility testing. Laboratory diagnostics are free of charge at the TH, and with a reduced charge at the NTH [15–17,20].

### Data collection and management

Data were prospectively collected for 10 years from 2008 to 2017. The data collection process is described in detail earlier [15–17,20]. In brief, pre-trained nurses filled the specifically designed forms, which included information like inpatient's name, age and sex; admission and discharge dates; department number; diagnosis determined by the consultant; when the surgery was performed and culture and susceptibility test was done; details about prescribed antibiotic(s) (generic and trade name, type of formulation, route of administration, dose, frequency and duration); and treatment outcome. The information was recorded for each patient admitted to the orthopedic wards in the two hospitals, for the entire period of their hospital stay. Patients who stayed in the hospital for at least one night and were above 15 years of age were considered as inpatients in the present study and included in the analyses.

Inpatients were compared based on the demographic variables, duration of hospital stay, type of surgery performed, indications (diagnoses), prescribed antibiotics, performed culture and antibiotic susceptibility test, duration of antibiotic treatment and the final treatment outcome. For meticulous analysis, the inpatients were divided into two groups, operated and non-operated. The operated inpatients are recommended to receive antibiotic prophylaxis to limit the perioperative sepsis; whereas prescribing antibiotics to non-operated inpatients depends

on the associated conditions and additional infectious diagnoses or presence of the symptoms of infection [21]. The most common indications for antibiotic prescribing in these two groups of inpatients were analyzed and presented. Moreover, the orthopedic infectious diagnoses, i.e., tuberculosis of bones and joints; cutaneous abscess, furuncle and carbuncle; cellulitis; osteomyelitis; diabetic foot, as well as other infectious diagnoses were grouped and presented as all indicated infectious diagnoses. Furthermore, the number of inpatients with multiple fractures was analyzed and presented, as these inpatients are more likely to have more complex surgeries, treatment and rehabilitation process and thus postponed recovery [22]. In the case of an inpatient having two or more diagnoses, the most relevant diagnosis for admission to the orthopedic department was considered during analyses.

Prescribed antibiotics were categorized according to the WHO Anatomical Therapeutic Chemical (ATC) classification and generic names of antibiotics [23]. The amount of total prescribed antibiotics was calculated by using Defined Daily Doses (DDDs) [23]. Since DDDs are available only for adults [24], patients below 15 years of age were excluded from the analysis (1750; TH-1217; NTH-533). January 2020 version of the ATC/DDD Index developed by the WHO Collaborating Centre for Drug Statistics Methodology (WHOCC) was used for the analysis [23]. DDDs were standardized to 1000 patient-days to enable comparison between the two hospitals. Mean values of DDDs for total prescribed antibiotics were calculated and compared between the two study hospitals [24]. The comparison was based on the presence of prescribed antibiotics in the latest available versions of the National List of Essential Medicines India (NLEMI, 2015) [25] and the WHO Model List of Essential Medicines (WHOMLEM, 2019) [26]. For the trend analysis, the measure DDD/1000 patient-days represents total antibiotic prescribing in orthopedic departments in each month over 10 years. Overall adherence to antibiotic prescribing guidelines was analyzed using following indicators: adherence to the NLEMI and the WHOMLEM; culture and susceptibility tests performed; prescriptions made using generic names; mean of DDD and DDD/1000 patient-days; operated vs. non-operated inpatients who were prescribed antibiotics. Finally, the analysis of temporal trends in each hospital provided additional justification for the antibiotic prescribing, by assessing if the percentage of prescribed antibiotics was supported by the percentage of inpatients, and if the percentage of the most frequently prescribed antibiotic class was supported by the percentage of the most common orthopedic diagnosis. Other possible reasons for the trends of antibiotic prescription were also investigated, such as number of performed operations and length of hospital stay. Percentages were calculated by dividing the monthly number by the yearly number of inpatients/ antibiotics/diagnoses.

## Statistical analysis

Mean, median and standard deviations were calculated for continuous variables and they were compared by Student's t-test after checking for normal distribution in each hospital. Frequency and percentages were calculated for categorical variables and they were compared by Pearson's chi-squared test. The trends of antibiotic use were analyzed using time series analysis. Linear regression was used to explain and evaluate the changes in trends of antibiotic use over time. Linear trend by month is given by coefficient ($\beta$), which is defined as the slope of the response over time. P-values <0.05 were considered statistically significant. Data were analyzed using Excel and STATA version 15.1 (Stata Corp., College Station, TX, USA).

The study was approved by the Ethics committee of Ruxmaniben Deepchand Gardi Medical College, Ujjain, with approval letter number 41-2/2007, 114/2010 and 311/2013. Being an observational study, the data collection did not interfere with the treatment or cause any additional risks for the patients. The patients were neither identified individually nor contacted by

the researchers at any stage of the study; therefore, the institutional ethics committee waived the requirement for obtaining an informed consent from the patients. Also, unique ID codes were assigned to all inpatients and the data were analyzed anonymously at the group level.

## Results

During 10 years, a total of 10,865 patients were admitted to both study hospitals; of those 6,446 were in the TH and 4,397 in the NTH. Table 1 presents the general characteristics of inpatients. In the TH, fewer inpatients were operated (23%) compared to the NTH (27%).

Table 2 shows that a total of 113760 antibiotic prescriptions were written for 6989 inpatients in both hospitals; 80% of antibiotics were prescribed in the TH, and 20% in the NTH. Adherence to the WHOMLEM was higher than adherence to the NLEMI in both hospitals. Thirty-five percent of the prescriptions in the TH and 40% in the NTH were fixed-dose combinations (FDCs) which had ATC codes assigned by WHOCC. The most commonly prescribed FDCs were ceftriaxone and β-lactamase inhibitor (J01DD63) (TH-18%; NTH-21%); cefoperazone and β-lactamase inhibitor (J01DD62) (TH-8%; NTH-4%); ceftazidime and β-lactamase

**Table 1. Characteristics of the inpatients at orthopedic departments in the teaching and the non-teaching hospital in Central India.**

| Characteristics of the inpatients | Teaching hospital | Non-teaching hospital | P-value |
|---|---|---|---|
| | n = 6,446 | n = 4,397 | |
| | N (%) | N (%) | $\chi^2$ test |
| **Sex** | | | |
| Male | 4,214 (65) | 3,115 (71) | 0.009 |
| Female | 2,232 (35) | 1,282 (29) | <0.001 |
| **Age** | | | |
| 15–30 | 1,857 (29) | 1,398 (32) | <0.001 |
| 31–45 | 1,926 (30) | 1,332 (30) | 0.071 |
| 46–60 | 1,527 (24) | 952 (22) | 0.486 |
| >60 | 1,124 (17) | 699 (16) | 0.510 |
| Missing age information | 12 | 16 | |
| **Treatment procedure** | | | |
| Operated | 1,479 (23) | 1,196 (27) | <0.001 |
| Prescribed antibiotics | 3,419 (53) | 3,570 (81) | <0.001 |
| Performed culture and susceptibility tests | 164 (3) | 17 (0) | <0.001 |
| **Outcome** | | | |
| Discharged from the hospital | 4,484 (69) | 3,758 (86) | <0.001 |
| Shifted within hospital to other wards | 53 (1) | 408 (9) | <0.001 |
| Absconded from the ward | 1,155 (18) | 139 (3) | <0.001 |
| Discharged on request | 749 (12) | 84 (2) | <0.001 |
| Referred to other hospital for further treatment | 1 (0) | 4 (0) | 0.072 |
| Died | 4 (0) | 4 (0) | 0.586 |
| | | | T-test |
| Length of hospital stay in days, Mean (SD) | 12.9 (11.4) | 5.8 (6.0) | <0.001 |
| Length of hospital stay in days, Median | 9 | 4 | |
| Length of antibiotic treatment in hospital in days, Mean (SD) | 7.1 (10.1) | 4.5 (4.6) | <0.001 |
| Length of antibiotic treatment in hospital in days, Median | 3 | 3 | |

Abbreviations: n, number of inpatients; $\chi^2$ test, utilizes number of inpatients; SD, standard deviation.

**Table 2. Antibiotic prescription details and adherence to the essential medicines lists at orthopedic departments in the teaching and the non-teaching hospital in Central India.**

| Variables | Teaching hospital | Non-teaching hospital | P-value |
|---|---|---|---|
| | n = 90,626 | n = 23,134 | |
| Antibiotic prescriptions adherent to the NLEMI, n (%) | 27,798 (31) | 7,819 (34) | <0.001[1] |
| Antibiotic prescriptions adherent to the WHOMLEM, n (%) | 58,798 (65) | 14,328 (62) | <0.001[1] |
| Prescribed FDCs listed by WHOCC[3], n (%) | 31,730 (35) | 9,194 (40) | <0.001[1] |
| Prescribed FDCs not listed by WHOCC, n (%) | 54 (0) | 73 (0) | <0.001[1] |
| Antibiotic prescriptions by generic name, n (%) | 33,962 (38) | 293 (1) | <0.001[1] |
| Prescribed DDD, mean (SD) | 0.9 (0.7) | 0.9 (0.9) | 1.000[2] |

[1] Pearson's chi-squared test was used for comparison of the variables.

[2] Student's t-test was used for comparison of the variables.

[3] Included in January 2020 list by WHOCC for Drug Statistics Methodology.

Abbreviations: n, number of prescriptions; NLEMI, National List of Essential Medicines in India (2015); WHOMLEM, World Health Organization Model List of Essential Medicines (2019); FDCs, fixed-dose combinations; WHOCC, World Health Organization Collaborating Centre for Drug Statistics Methodology; DDD, defined daily dose; SD, standard deviation.

inhibitor (J01DD52) (NTH-7%); and piperacillin and β-lactamase inhibitor (J01CR05) (NTH-4%). In both hospitals, very few FDCs (TH-54; NTH-73) were prescibed without WHOC-C-ATC codes. Antibiotics were prescribed by generic name in 38% prescriptions in the TH and one percent in the NTH. The mean value of prescribed DDDs in both hospitals was less than the recommended value- one.

In both hospitals, other β-lactams (J01D) (TH-39%; NTH-75%), followed by aminoglycosides (J01G) (TH-35%; NTH-10%) were the most frequently prescribed antibiotic classes. From other β-lactams, most commonly prescribed antibiotic subgroup in both hospitals were 3rd generation cephalosporins (J01DD) (TH-39%; NTH-65%).

The most common orthopedic indications in both hospitals were fractures of spine and limbs (TH-48%; NTH-48%), followed by dorsalgia (TH-11%; NTH-3%). In the TH, 854 (13%) inpatients had two or more diagnoses; whereas, in the NTH, 462 (11%) inpatients had two or more diagnoses, e.g. fracture of humerus and chronic osteomyelitis, fracture of femur and diabetes mellitus etc. Infectious indications accounted for 7% of all the diagnoses in the TH and 12% of all the diagnoses in the NTH. Multiple fractures accounted for three percent of diagnoses in each hospital. Majority of the operated inpatients were prescribed antibiotic prophylaxis (TH-99%; NTH-97%) (Table 3). In non-operated inpatients, infectious diagnoses accounted for eight percent of diagnoses in the TH and 14% of diagnoses in the NTH, whereas multiple fractures accounted for two percent of non-operated diagnoses in both hospitals. Antibiotics were prescribed to 40% of non-operated inpatients in the TH and 75% in the NTH (Table 3).

## Time series analysis

In the TH, the value of DDD/1000 patient-days showed an increasing trend (β = 3.23, p<0.001) over 10 years; with the range of 225 to 6429, and mean 2658 (±529) DDD/1000 patient-days (Fig 1A). Similarly, in the NTH, the trend of DDD/1000 patient-days increased (β = 1.02, p<0.001), but within the lower range; 56–387, and mean 162 (±69) DDD/1000 patient-days (Fig 1B). In both hospitals, the level of prescribed antibiotics was in line with the level of inpatients on monthly basis and significantly increased over time (TH-β = 0.01, p<0.001, Fig 2A and NTH-β = 0.03, p<0.001, Fig 2B). Likewise, in both hospitals, the level of fractures and the level of prescribed 3rd generation cephalosporins had the same increasing trend (TH-β =

**Table 3. Comparison of numbers of operated/non-operated inpatients who were prescribed antibiotics with respect to the most common diagnoses at orthopedic departments in the teaching and the non-teaching hospital in Central India.**

| | OPERATED | | | | NON-OPERATED | | | |
|---|---|---|---|---|---|---|---|---|
| | Teaching hospital | | Non-teaching hospital | | Teaching hospital | | Non-teaching hospital | |
| **Total inpatients, N (%)** | N = 1479 | | N = 1196 | | N = 4967 | | N = 3201[#] | |
| | Frequency of diagnosis | Inpatients prescribed antibiotic | Frequency of diagnosis | Inpatients prescribed antibiotic | Frequency of diagnosis | Inpatients prescribed antibiotic | Frequency of diagnosis | Inpatients prescribed antibiotic |
| | n | n* (%) | n | n* (%) | n | n* (%) | n | n* (%) |
| | | 1458 (99) | | 1155 (97) | | 1961 (40) | | 2415 (75) |
| *ICD-10 Codes and Diagnoses* | | | | | | | | |
| M 51 Other intervertebral disc disorders | 63 | 53 (84) | 22 | 21 (95)[#] | 618 | 82 (13) | 73 | 44 (60) |
| M 54 Dorsalgia | 7 | 5 (71) | 4 | 4 (100) | 693 | 72 (10) | 134 | 52 (39) |
| S 32–S 82 Fractures of spine and limbs | | | | | | | | |
| *S 32 lumbar spine and pelvis* | 16 | 16 (100) | 15 | 15 (100) | 108 | 33 (31) | 113 | 87 (77)[#] |
| *S 42 shoulder and upper arm* | 98 | 98 (100) | 9 | 92 (97) | 280 | 138 (49) | 150 | 119 (79)[#] |
| *S 52 forearm* | 154 | 154 (100) | 108 | 101 (94) | 344 | 183 (53) | 142 | 110 (77) |
| *S 62 wrist and hand level* | 42 | 41 (98) | 54 | 51 (94)[#] | 67 | 27 (40) | 102 | 72 (71)[#] |
| *S 72 femur* | 381 | 380 (100) | 207 | 206 (100)[#] | 835 | 456 (55) | 414 | 346 (84)[#] |
| *S 82 lower leg, including ankle* | 269 | 269 (100) | 275 | 271 (99)[#] | 500 | 304 (61) | 440 | 369 (84)[#] |
| T 14 Injury of unspecified body region | 59 | 59 (100) | 39 | 38 (97) | 111 | 44 (40) | 79 | 61 (77)[#] |
| Multiple fractures | 98 | 98 (100) | 36 | 36 (100)[#] | 122 | 71 (58) | 76 | 66 (87)[#] |
| All bacterial infectious diagnoses | 68 | 67 (99) | 74 | 73 (99) | 395 | 192 (49) | 455 | 359 (79)[#] |
| Other non-infectious diagnoses** | 218 | 218 (100) | 259 | 247 (95)[#] | 857 | 359 (42) | 1006 | 730 (73)[#] |

N = Total number of inpatients, n = frequency of diagnoses, n* = number of inpatients who were prescribed antibiotics. The percentage n* (%) is calculated for the number of inpatients who were prescribed antibiotic with specific diagnosis out of the total number of inpatients with that diagnosis.

**Includes illegible or missing diagnoses (TH-43; NTH-25).

[#] P-Value ($\chi^2$ test) is statistically significant.

Abbreviations: TH, teaching hospital; NTH, non-teaching hospital.

0.01, $p<0.001$, Fig 3A and NTH-$\beta$ = 0.03, $p<0.001$, Fig 3B) over 10 years. In both hospitals, there was a very slight increase in number of performed operations (TH-$\beta$ = 0.008, $p<0.001$ and NTH-$\beta$ = 0.006, $p<0.001$) and slight decrease in the length of hospital stay (TH-$\beta$ = -0.013, p = 0.004 and NTH-$\beta$ = -0.04, $p<0.001$) over 10 years.

## Discussion

To the best of our knowledge, this is the first cross-sectional study that presents trends of antibiotic prescribing over 10 years at orthopedic departments in a LMIC. Overall, the TH had more inpatients, longer median duration of hospital stay and more antibiotic prescriptions per

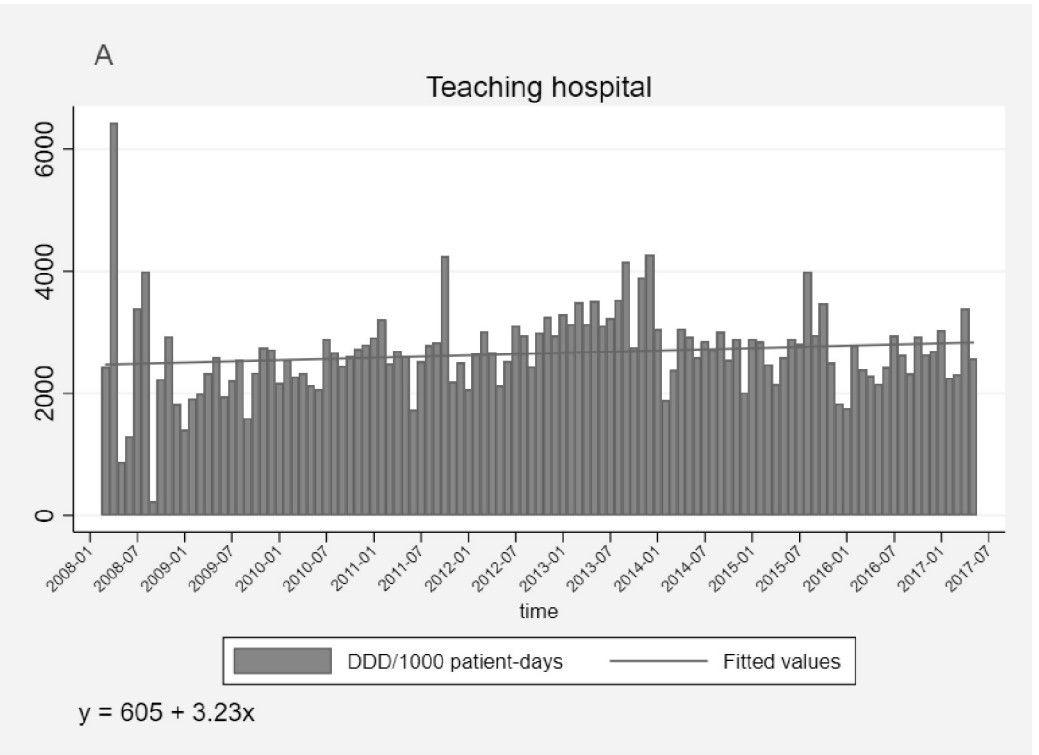

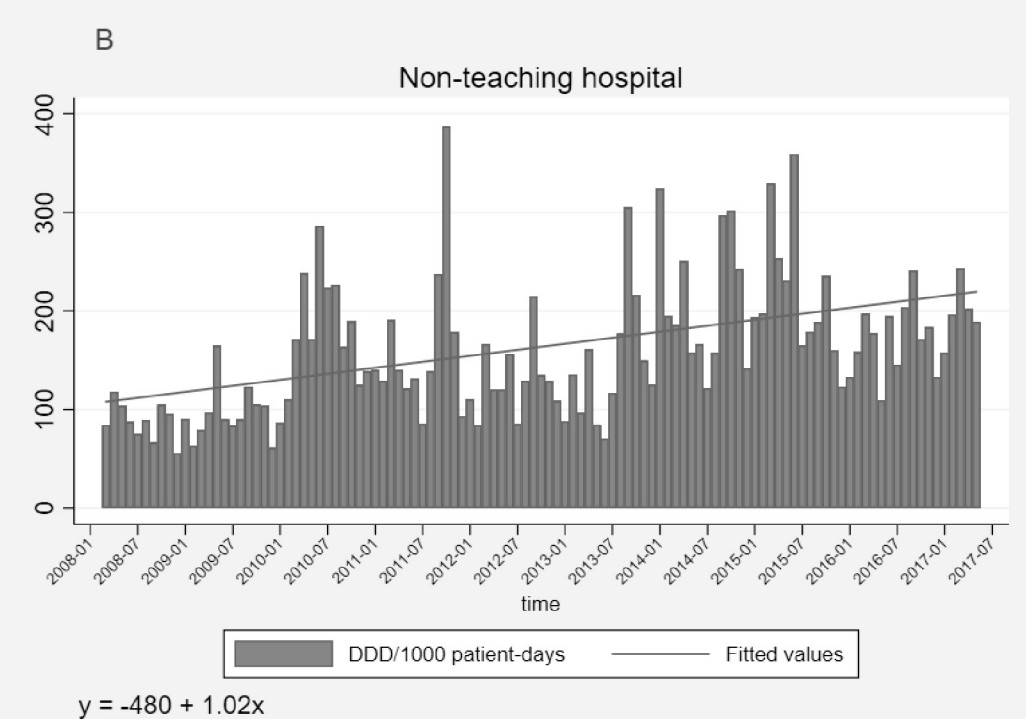

**Fig 1. DDD/1000 patient-days at the orthopedic departments of the teaching hospital (1A) and the non-teaching hospital (1B) in Central India over 10 years.** Abbreviations: DDD, defined daily dose.

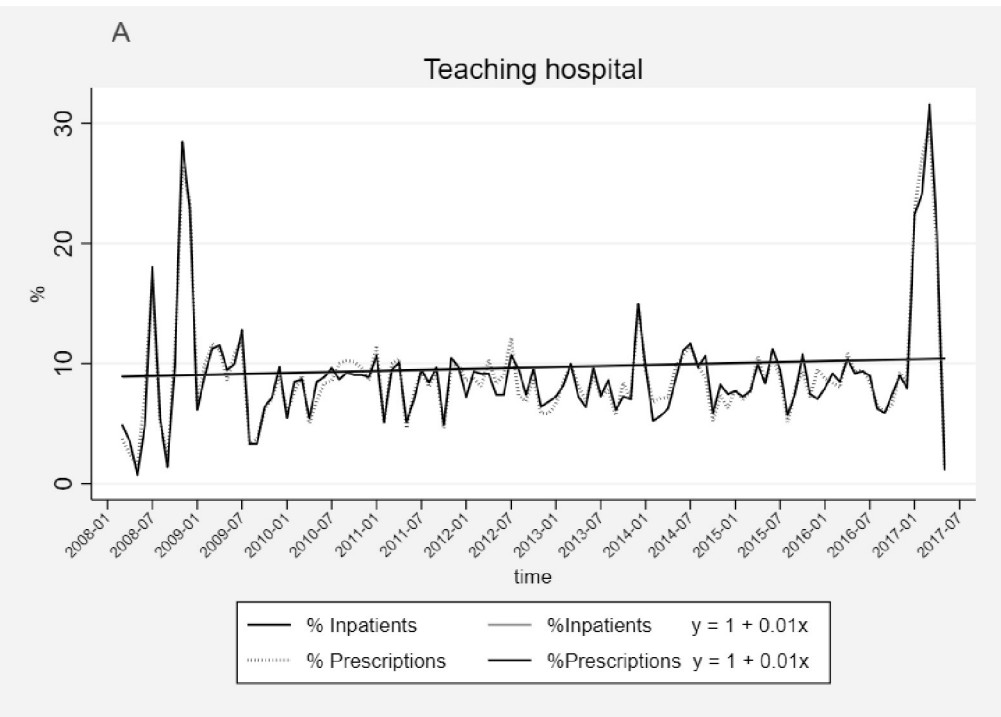

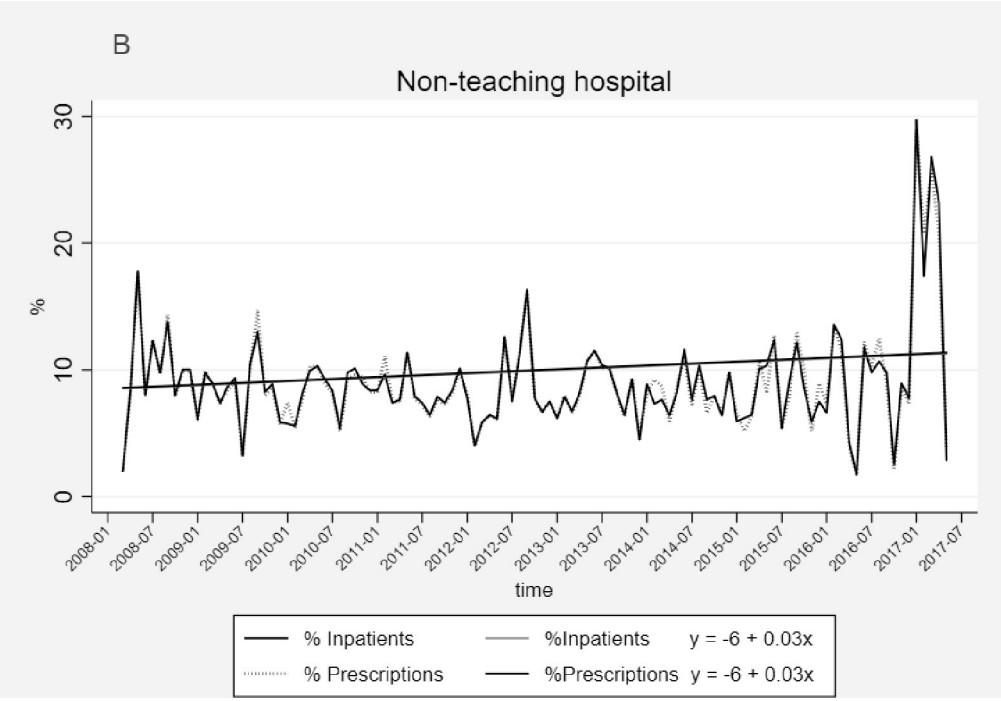

**Fig 2. Percentage of inpatients vs. percentage of antibiotic prescriptions at the orthopedic departments of the teaching hospital (2A) and the non-teaching hospital (2B) in Central India over 10 years.**

patient compared to the NTH. However, the proportions of operated inpatients and inpatients who were prescribed antibiotics were significantly higher in the NTH than in the TH. Prescriptions made at the NTH were significantly more adherent to the NLEMI (34%) compared to

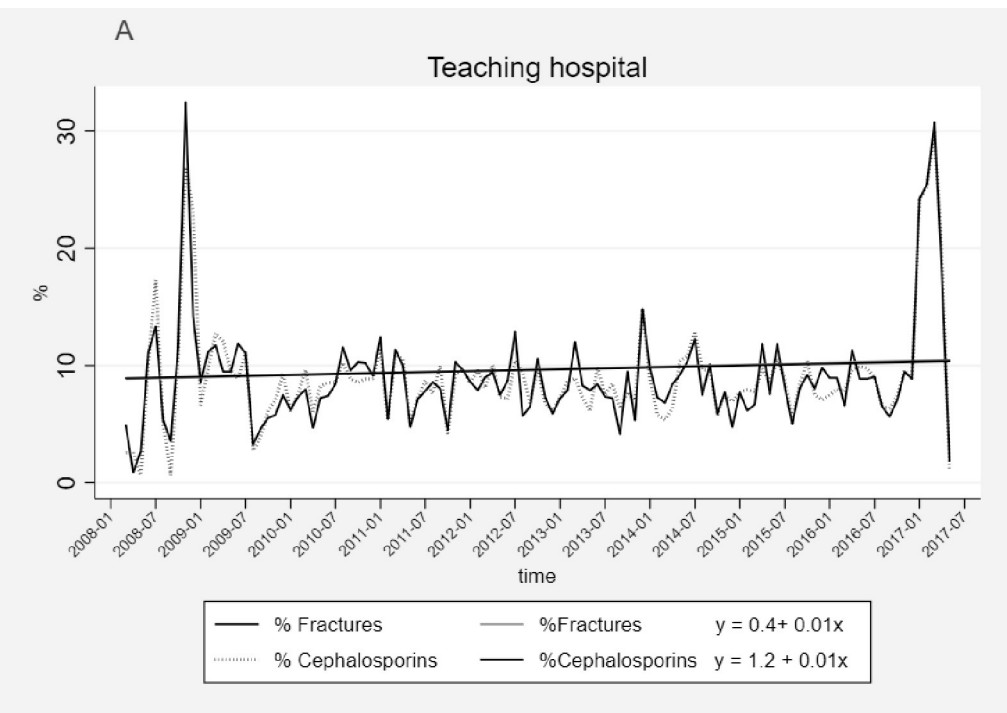

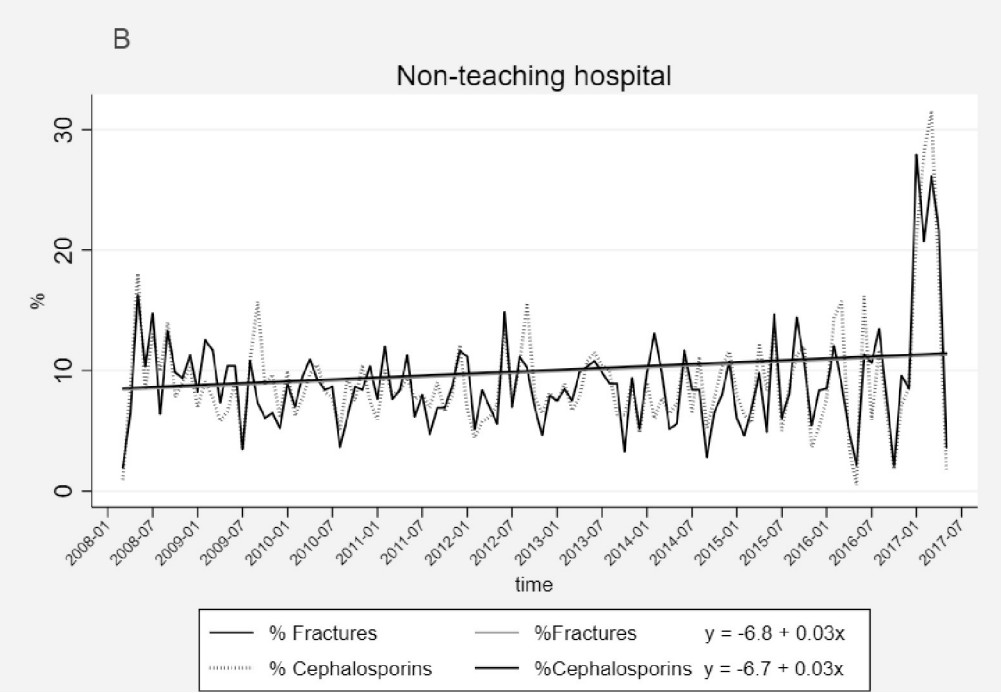

**Fig 3. Percentage of fractures vs. percentage of prescribed 3[rd] generation cephalosporins at the orthopaedic departments of the teaching hospital (3A) and the non-teaching hospital (3B) in Central India over 10 years.**

the TH (31%). However, the adherence to the WHOMLEM was statistically significantly higher in the TH (65%) than in the NTH (62%). The most prescribed antibiotic subclass in both hospitals were 3[rd] generation cephalosporins (J01DD). The prescriptions of FDCs with

assigned ATC codes by the WHOCC were more frequent in our study compared to the previous studies. The most common orthopedic indications in both hospitals were fractures of spine and limbs. Total antibiotic use per patient and per hospital days was 16 times higher in the TH compared to the NTH (TH-2658; NTH-162). Over 10 years, the trend of total antibiotic use increased significantly in both hospitals. This rate of increase was 3 times higher in the TH ($\beta$ = 3.23) compared to the rate in the NTH ($\beta$ = 1.02). In both hospitals, the number of prescribed antibiotics was supported by the number of inpatients on monthly basis over 10 years. Similarly, the increasing trend of the most frequently prescribed 3[rd] generation cephalosporins followed the increasing trend of the most common indications, fractures.

Previous studies, conducted in other departments of the two hospitals, showed more appropriate prescribing practices in the TH compared to the NTH [15–17]. This was partially similar in this study; prescribing by generic names was significantly higher in the orthopedic departments of the TH compared to the NTH, whereas prescribing of FDCs was significantly higher in the NTH than in the TH. Our results suggest that the prescription of FDCs with assigned ATC codes in both hospitals was higher than in the previous studies and thus, more appropriate. However, it is noteworthy that the most prescribed FDCs in our study were not listed by WHOCC until 2018 and they are still not recommended by neither the NLEMI nor the WHOMLEM, therefore, prescribing of those FDCs cannot be considered as rational. Moreover, our study confirms low orders of culture and antibiotic susceptibility testing in both hospitals, which had also been indicated in the previous studies [15–17].

There is a paucity of studies that investigated antibiotic prescribing patterns at orthopedic departments in LMICs [18,19]. Out of a few conducted, two recent studies from Gujarat, India and Larkana, Pakistan presented other β-lactams (J01D) as the most frequently prescribed antibiotic class; 41% and 33% respectively [18,19]. Antibiotics were prescribed by generic names in 22% of prescriptions in the private sector hospital in Gujarat [18]. However, these studies had a smaller sample size (200 patients) and shorter duration (up to 6 months) and thus do not substantiate the pattern of prescribing. On the other hand, the results of our study present not only antibiotic prescribing patterns but also antibiotic prescribing trends and confirm that the most frequently prescribed antibiotic class at orthopedic departments were other β-lactams, with higher numbers (TH-39%; NTH-75%) than in the study from Larkana [19]. Furthermore, the percentage of antibiotic prescriptions by generic name was higher in the TH (38%) and substantially lower in the NTH (1%) compared with the results of the study in Gujarat [18]. The two study hospitals presented two opposites in the results.

Fractures of spine and limbs were the most common indications in both study hospitals. However, it is noteworthy that fractures are not necessarily indications for prescription of antibiotics, unless: a) there is a presence of open or multiple fractures that present a greater risk for infection; b) there is associated infectious diagnosis; and/or, c) there is a need for surgery [21]. Majority of the operated patients in our study were prescribed antibiotics as prophylaxis, according to the recommendations for surgeries to minimize perioperative infections [21]. Also, a considerable proportion of non-operated inpatients were prescribed antibiotics (TH-40%; NTH-75%), while all infectious diagnoses and multiple fractures accounted only for a small proportion of diagnoses (TH-10%; NTH-16%).

Empiric prescribing based on the clinical diagnoses might be the possible reason for prescribing antibiotics to the non-operated inpatients without any clear infectious indication. The possibility of missing detailed information about indications of an infection, such as type of fracture (open/closed) and symptoms of infection (swelling, warmth and redness around the wound, etc.), cannot be excluded in some cases. Thus, the exact number of infectious diagnoses was most likely higher than presented. Furthermore, our findings also suggest that antibiotic treatment patterns did not always match the orthopedic indications. The type of

prescribed antibiotics for prevention or treatment of infections in orthopedic surgeries acknowledged the recommendations; in general, other β-lactams (cephalosporins) and aminoglycosides [10,11,14] were the most commonly prescribed antibiotic classes for such cases. The results of the time series analysis infer that antibiotic prescription in both hospitals was supported by the number of inpatients.

Overall adherence to the essential medicine lists and guidelines in both hospitals is a point of discussion. Firstly, it was expected that the overall adherence to the NLEMI would be higher than adherence to the WHOMLEM because the NLEMI is nationally contextualized for India; however, our results show that adherence to the WHOMLEM was higher in both hospitals than to the NLEMI. This might be because more antibiotics are listed in the WHOMLEM than in the NLEMI. However, in-depth study is recommended to understand all underlying reasons for the differences in adherences to the two essential medicine lists. Next highlight is that the antibiotics were mostly prescribed by trade name in both hospitals, though the standardized and cost-effective way of prescribing is by generic name, which is also recommended by the government of India [15–17]. Previous studies conducted in both hospitals suggested that higher prescription by generic name in the TH could be due to two possible main reasons. Firstly, in the TH, drugs are purchased centrally by the hospital management, to assure the purchase of generic drugs that are equally safe and effective yet lower cost, compared to the branded drugs [15–17]. Secondly, in the NTH, there was no restriction on the consultants to communicate with the pharmaceutical sales representatives. Many studies worldwide have reported that the pharmaceutical sales representatives influence the prescribers to write prescriptions of certain brands of drugs. Visits made by these representatives can, therefore, increase the trade name prescribing [15–17].

Due to the paucity of studies about antibiotic prescribing trends over time specifically in orthopedic departments, we compared our results with the results of the study about antibiotic prescribing trends over 10 years in one tertiary hospital in South Korea [27]. Mean antibiotic prescription was 3 times lower (921 DDD/1,000 patient-days) in South Korean hospital than in the TH, and 6 times higher than mean antibiotic prescription in the NTH. The most commonly prescribed antibiotic class in South Korean and our study hospitals was the same; 3rd generation cephalosporins [27]; however, the frequency of prescribing was lower (19%) in South Korean study compared to the results of our study (TH-39%; NTH-65%). The prescription of 3rd generation cephalosporins showed a significant decreasing trend ($\beta = -0.295$) in South Korean hospital [27], while in both of our study hospitals, the trend of the use of 3rd generation cephalosporins increased significantly (TH-$\beta = 0.01$; NTH-$\beta = 0.03$).

General increase in rate of prescription of 3rd generation cephalosporins in both hospitals followed the increase in number of fracture cases over 10 years. Additionally, the increase in prescriptions of 3rd generation cephalosporins could be explained by slight increase in performed operations, although the rates of increase of antibiotic prescriptions in both hospitals were 1.3 and 5 times higher (TH-$\beta = 0.01$; NTH-$\beta = 0.03$) than the rates of increase in the performed operations (TH-$\beta = 0.008$; NTH-$\beta = 0.006$). Furthermore, the length of hospital stay slightly decreased in both hospitals over 10 years, so this did not support the increase in prescriptions of 3rd generation cephalosporins.

Furthermore, mean DDD value was lower than the ideal value 1 in both hospitals [24]. The lower value of DDD could be due to change of DDDs over 10 years of the study period. We have used the most recent 2020 ATC/DDD index for analysis [23], while the data were collected until 2017. Finally, our results demonstrated that antibiotic prescribing in both hospitals was mostly empirical; since to date, none of the hospitals has implemented antibiotic prescription guidelines in the orthopedic departments. There is a need to study further the reasons for continuous empiric treatment and underutilization of laboratory services, low adherence to

the guidelines and overprescribing of antibiotics in orthopedics departments. We suggest conducting qualitative studies in the form of personal interviews with the consultants to highlight the areas for intervention, with the aim to enable higher prescription of narrow-spectrum antibiotics.

## Methodological considerations

The study is of long duration; 10 years of prospective data collection of all inpatients admitted to orthopedic departments ensured a relatively big and representative sample. Thus, our findings regarding the most common orthopedic indications and the most frequently prescribed antibiotic classes might be generalizable to India and other similar settings globally. Furthermore, nurses, who were specifically trained for data collection, used the same detailed form for data collection in both hospitals and thus enabled a good baseline for comparison of the two hospitals. In addition, this study used ATC/DDD system, which is the best available method for comparison of antibiotic use between health care settings, regions and countries; as well as trends in antibiotic use over time [23]. However, this method does not take into account that the recommended dose can differ according to the age, indication and severity of disease; as only one DDD is given per generic substance [23]. Consequently, the pediatric patient group was excluded from the analyses. Furthermore, DDDs can change over time which can cause difficulties in comparison of patterns and trends in antibiotic use. This was partly solved by using the latest available 2020 ATC/DDD Index. In addition, there is a possibility of missing data due to the manual data collection by nurses. The diagnoses were not validated externally, as this was beyond our aim. A complete picture would have been provided if each antibiotic prescription had been accompanied by culture and susceptibility test. Additionally, the inclusion of information about the type of fracture and symptoms of infection, for diagnoses that are not clinically infectious, could have helped more to assess the rationality of antibiotic prescribing. Nevertheless, concluding about the rationality of antibiotic prescribing was not the aim of this study. A further limitation of the study is that the information about the surgery status of inpatients and preformed culture and susceptibility tests started to be collected from 2011, roughly three years after the beginning of the study. Therefore, true numbers of operated inpatients and culture tests are probably higher; hence the use of antibiotics in those cases would have been justified. Finally, the choice of the linear regression model for the statistical analysis is a suboptimal option, since linear regression assumes that the data are independent of each other, while in the time series analysis, the variables of interest are dependent on time [28]. However, in this case, the rationale for the use of time series analysis was to explain the existing trend and not to predict the future trend; therefore, the use of the linear regression model was justified.

## Conclusion

In both hospitals, adherence to lists of essential medicines was inadequate. The most frequently prescribed antibiotic class were 3[rd] generation cephalosporins. Majority of operated inpatients were prescribed antibiotic prophylaxis, according to the recommended practice in orthopedic surgery. However, a substantial number of non-operated inpatients were also prescribed antibiotics without a clear indication for antibiotic prescription. In both hospitals, there was an increasing trend of antibiotic use over 10 years, with higher total antibiotic use and rate of increase in the TH compared to the NTH. The results of time series analysis indicate that antibiotic use was justified by the number of inpatients. Antibiotic prescribing in both hospitals was mostly empirical.

This study showed the need to improve antibiotic prescribing in orthopedic departments of the two study hospitals, starting from the availability of antibiotic prescribing guidelines and more frequent culture and susceptibility testing. Also, the need for development of orthopedic indication-specific antibiotic prescribing guidelines, tailored specifically to local resistance patterns, is underlined. More frequent update of the NLEMI is proposed, as it is done with the WHOMLEM.

## Supporting information

**S1 Fig. Percentages of four most prescribed antibiotic classes at the orthopaedic departments of the teaching (4A) and the non-teaching hospital (4B) in Central India over 10 years.**
(TIF)

**S2 Fig. Percentages of four most prescribed antibiotic substances at the orthopaedic departments of the teaching (5A) and the non-teaching hospital (5B) in Central India over 10 years.**
(TIF)

**S1 Appendix. Appendix A: Comparison of prescription trends of four most prescribed antibiotic classes.**
(DOCX)

**S2 Appendix. Appendix B: Comparison of prescription trends of the four most frequently prescribed antibiotic substances.**
(DOCX)

## Acknowledgments

We would like to extend our thanks to the Medical Director, Dr. V. K. Mahadik and the management of both study settings for providing the permission and support throughout the study. We would like to thank all nursing staff and orthopedic consultants working at C. R. Gardi Hospital, R. D. Gardi Medical College in Surasa, and Ujjain Charitable Trust Hospital and Research Center, Ujjain, India, for their cooperation and help.

## Author Contributions

**Conceptualization:** Cecilia Stalsby-Lundborg, Megha Sharma.

**Data curation:** Megha Sharma.

**Formal analysis:** Kristina Skender, Megha Sharma.

**Funding acquisition:** Cecilia Stalsby-Lundborg.

**Investigation:** Vivek Singh, Megha Sharma.

**Methodology:** Megha Sharma.

**Project administration:** Megha Sharma.

**Resources:** Megha Sharma.

**Software:** Cecilia Stalsby-Lundborg.

**Supervision:** Megha Sharma.

**Validation:** Vivek Singh, Megha Sharma.

**Visualization:** Kristina Skender, Cecilia Stalsby-Lundborg, Megha Sharma.

**Writing – original draft:** Kristina Skender, Megha Sharma.

**Writing – review & editing:** Kristina Skender, Vivek Singh, Cecilia Stalsby-Lundborg, Megha Sharma.

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
