## [Decision Letter · Decision Letter 0]

26 Nov 2020

PONE-D-20-33344

Trends and patterns of antibiotic prescribing at orthopedic inpatient departments of two private-sector hospitals in Central India: A 10-year observational study

PLOS ONE

Dear Dr. Sharma,

Thank you for submitting your manuscript to PLOS ONE. After careful consideration, we feel that it has merit but does not fully meet PLOS ONE’s publication criteria as it currently stands. Therefore, we invite you to submit a revised version of the manuscript that addresses the points raised during the review process.

Possible explanations for some of the observations would be of value.

We look forward to receiving your revised manuscript.

Kind regards,

Iddya Karunasagar

Academic Editor

PLOS ONE

Additional Editor Comments:

Two reviewers have commented on the manuscript and have raised some very pertinent points for improving the manuscript. Please address all reviewer comments point by point.

Journal Requirements:

"The study was funded by the Swedish Research

 Council (K2007-70X-20514-01-3, K2010–396 70X-20514-04-3, 2017–01327) and Asia Link (348–

2006-6633). KS is the recipient of scholarships by Karolinska Institutet Foundation and Elisabeth and

Alfred Ahlqvist's Foundation from Apotekarsocieteten, Sweden. MS is the recipient of Erasmus Mundus

Lot-15. The funding agencies had no role in the study plan, design, data collection and analysis,

interpretation of the results, decision to publish, or preparation of the manuscript.

We declare no conflict of interest."

Reviewers' comments:

Reviewer's Responses to Questions

**Comments to the Author**

1. Is the manuscript technically sound, and do the data support the conclusions?

Reviewer #1: Yes

Reviewer #2: Yes

2. Has the statistical analysis been performed appropriately and rigorously? 

Reviewer #1: Yes

Reviewer #2: Yes

3. Have the authors made all data underlying the findings in their manuscript fully available?

Reviewer #1: Yes

Reviewer #2: No

4. Is the manuscript presented in an intelligible fashion and written in standard English?

Reviewer #1: Yes

Reviewer #2: Yes

5. Review Comments to the Author

Reviewer #1: The paper " Trends and patterns in antibiotic prescribing at Orthopaedic in patient departments at two private sector hospitals in Central India: A 10 year observational study

The paper is well written with an eye to detail.

Observation :

1. The first two paragraphs of the introduction section could be shortened to make the purpose of the paper more toned and meaningful.

Reviewer #2: Reviewer Comments PONE-D-20-33344

Trends and patterns of antibiotic prescribing at orthopedic inpatient departments of two

Private-sector hospitals in Central India: A 10-year observational study

Authors have studied an important area of antibiotic prescribing in orthopaedic in-patients in two private set ups in India. My specific comments are as follows

1. Page 9 Line 138: The comparison was based on the presence of prescribed antibiotics in the latest available versions of the National List of Essential Medicines India (NLEMI, 2015) and the WHO Model List of Essential Medicines (WHOMLEM, 141 2019).

Line 179: Adherence to the WHOMLEM was higher than adherence to the NLEMI in both hospitals.

There is a difference in the adherence. What may be the causes?

2. There is a wide difference in antibiotics prescribed by generic name and brand name between the 2 hospitals. What is the reason?

3. Line 272: However, it is noteworthy that fractures are not necessarily indications for prescription of antibiotics, unless: a) there is a presence of open or multiple fractures that present a greater risk for infection; b) there is associated infectious diagnosis;

Is there a break up of these a, and b, in the two hospital? OR is there a difference in the study participants between the two hospitals, as far as diagnosis is compared.

4. 232: first repeated – spelling mistake? May be it should be ‘first reported’

5. It will be a better idea to classify antibiotic usage as per “AWARE” category-Access, Watch and Restricted class. This will give an idea which class of drugs were trending.

Is there increase in Access class/ Watch class/Restricted class of antibiotics?

6. There is a general increase in antibiotic prescription in TH and NTH; But the rate of change is higher in NTH.

This change is similar to increase in number of fracture cases.

However,

Is there an increase in operations performed?/ or is there any trend?

Is there an increase in length of hospital stay/ or is there any trend?

Is there any change in mortality/survival?

6. PLOS authors have the option to publish the peer review history of their article (what does this mean?). If published, this will include your full peer review and any attached files.

Reviewer #1: No

Reviewer #2: **Yes: **Kavitha Saravu

---

## [Author Response · Author response to Decision Letter 0]

30 Dec 2020

Reviewer Comments PONE-D-20-33344

Trends and patterns of antibiotic prescribing at orthopedic inpatient departments of two

private-sector hospitals in Central India: A 10-year observational study 

Reviewer #1: 

The first two paragraphs of the introduction section could be shortened to make the purpose of the paper more toned and meaningful.

Authors’ Response: Thank you for this suggestion. We have now modified and shortened this part of introduction on page 5, line 51, as suggested.

“Antibiotic resistance is a serious threat to public health globally and it extensively affects health and economy specifically of low- and middle-income countries (LMICs), including India (5,6). It is estimated that by 2050, antibiotic resistance will cause 10 million deaths per year worldwide, out of which 2 million deaths are projected in India (7). India is one of the biggest consumers of antibiotics in the world, and despite the worldwide decrease in infectious diseases, antibiotic use in India is on the rise (8). From 2000 to 2015, antibiotic consumption in India increased by 103%, which is more than in any other country; due to remaining infectious disease burden, overall increased access to antibiotics and misuse (8).”

REFERENCES:

5. Chandy SJ, Naik GS, Balaji V, Jeyaseelan V, Thomas K, Lundborg CS. High cost burden and health consequences of antibiotic resistance: the price to pay. J Infect Dev Ctries. 2014 Sep 12;8(09):1096–102. 

6. Gandra S, Tseng KK, Arora A, Bhowmik B, Robinson ML, Panigrahi B, et al. The Mortality Burden of Multidrug-resistant Pathogens in India: A Retrospective, Observational Study. Clin Infect Dis. 2019 Aug 1;69(4):563–70. 

7. Dixit A, Kumar N, Kumar S, Trigun V. Antimicrobial Resistance: Progress in the Decade since Emergence of New Delhi Metallo-β-Lactamase in India. Indian J Community Med Off Publ Indian Assoc Prev Soc Med. 2019 Mar;44(1):4–8. 

8. Klein EY, Van Boeckel TP, Martinez EM, Pant S, Gandra S, Levin SA, et al. Global increase and geographic convergence in antibiotic consumption between 2000 and 2015. Proc Natl Acad Sci. 2018 Apr 10;115(15):E3463–70. 

Reviewer #2: 

Authors have studied an important area of antibiotic prescribing in orthopaedic in-patients in two private set ups in India. My specific comments are as follows

1. Page 9 Line 138: The comparison was based on the presence of prescribed antibiotics in the latest available versions of the National List of Essential Medicines India (NLEMI, 2015) and the WHO Model List of Essential Medicines (WHOMLEM, 141 2019).

Line 179: Adherence to the WHOMLEM was higher than adherence to the NLEMI in both hospitals. 

There is a difference in the adherence. What may be the causes?

Authors’ Response: Thank you for this question. We agree that it is important to discuss this point. We have modified the text related to the possible reasons for differences in the adherence in the discussion section on page 18, line 298. We have now elaborated it as per your suggestion. 

“Firstly, it was expected that the overall adherence to the NLEMI would be higher than adherence to the WHOMLEM because the NLEMI is nationally contextualized for India; however, our results show that adherence to the WHOMLEM was higher in both hospitals than to the NLEMI. This might be because more antibiotics are listed in the WHOMLEM than in the NLEMI. However, in-depth study is recommended to understand all underlying reasons for the differences in adherences to the two essential medicine lists.”

2. There is a wide difference in antibiotics prescribed by generic name and brand name between the 2 hospitals. What is the reason?

Authors’ Response: Thank you for acknowledging this point. We have highlighted this point in methods section on page 7, line 101. We have now modified the text, as below:

“Additionally, in the NTH, there is no administrative restriction on doctors, and they can be contacted by the pharmaceutical sales representatives. Furthermore, in the NTH the prescribed medicines could be bought from any of the pharmacies located in the area.”

In addition, we have added following text in the discussion section on page 18, line 305:

“Previous studies conducted in both hospitals suggested that higher prescription by generic name in the TH could be due to two possible main reasons. Firstly, in the TH, drugs are purchased centrally by the hospital management, to assure the purchase of generic drugs that are equally safe and effective yet lower cost, compared to the branded drugs (15-17). Secondly, in the NTH, there was no restriction on the consultants to communicate with the pharmaceutical sales representatives. Many studies worldwide have reported that the pharmaceutical sales representatives influence the prescribers to issue prescriptions of certain brands of drugs. Visits made by these representatives can therefore increase the trade name prescribing (15-17).”

REFERENCES: 

15. Sharma M, Damlin AL, Sharma A, Stålsby Lundborg C. Antibiotic prescribing in medical intensive care units – a comparison between two private sector hospitals in Central India. Infect Dis. 2015 May 4;47(5):302–9.

16. Sharma M, Damlin A, Pathak A, Stålsby Lundborg C. Antibiotic Prescribing among Pediatric Inpatients with Potential Infections in Two Private Sector Hospitals in Central India. Palaniyar N, editor. PLOS ONE. 2015 Nov 5;10(11):e0142317. 

17. Sharma M, Eriksson B, Marrone G, Dhaneria SP, Stålsby Lundborg C. Antibiotic prescribing in two private sector hospitals; one teaching and one non-teaching: A cross-sectional study in Ujjain, India. BMC Infect Dis. 2012 Dec;12(1):155.

Line 272: However, it is noteworthy that fractures are not necessarily indications for prescription of antibiotics, unless: a) there is a presence of open or multiple fractures that present a greater risk for infection; b) there is associated infectious diagnosis; 

Is there a break up of these a, and b, in the two hospital? OR is there a difference in the study participants between the two hospitals, as far as diagnosis is compared.

Authors’ Response: Thank you for this question. We have elaborated this point in results section on page 12, line 206:

“Infectious indications accounted for 7% of all the diagnoses in the TH and 12% of all the diagnoses in the NTH. Multiple fractures accounted for three percent of diagnoses in each hospital.”

In addition, the breakup of point a) and b), as well as difference/ comparison of diagnoses between the two hospitals is presented in Table 3 (page 14). However, the emphasis of present study was not on comparison of the diagnoses, but rather on the rationality of antibiotic prescription in the study hospitals. 

In Table 3, we emphasized firstly, on the division of operated and non-operated patients and secondly on comparing the patterns of prescribing antibiotics to the operated/ non-operated patients having the same diagnosis in the study hospitals. 

However, if we disregard the division between operated and non-operated patients and sum up all the patients with multiple fractures in each hospital, as well as all the patients with all bacterial infectious diagnoses in each hospital, the numbers are given as follows:

 TH NTH p-value

Total patients/ diagnoses n=6,446 n=4,397 Χ2 test

 n (%) n (%) 

Multiple fractures 220 (3.4) 112 (2.5) 0.013

All bacterial infectious diagnoses 463 (7.2) 529 (12.0) <0.001

From this calculation it is visible that multiple fractures are significantly higher in the TH but can be approximated to 3% of diagnoses in each hospital, while all bacterial infectious diagnoses are significantly higher in the NTH, as already mentioned on page 12, line 206.

3. 232: first repeated – spelling mistake? May be it should be ‘first reported’

Authors’ Response: Thank you for pointing to this spelling mistake. We have now corrected this in the main text (page 15, line 239). 

4. It will be a better idea to classify antibiotic usage as per “AWARE” category-Access, Watch and Restricted class. This will give an idea which class of drugs were trending.

Is there increase in Access class/ Watch class/Restricted class of antibiotics?

Authors’ Response: Thank you for this suggestion. We agree that the proposed ‘AWARE’ classification might give a different perspective to the results. However, the scope of this study was to analyze and present the overall patterns and trends of antibiotic prescriptions in the study settings, for the first time. Further, we have also cross-checked the trends with the indications, to study the justification of the prescribed antibiotics. 

Analysis of antibiotics as per “AWARE” classification might be interesting to analyze antibiotic usage based on “AWARE” classification to see the trends of each antibiotic class. Presently however, resources are not available for such an analysis. 

5. There is a general increase in antibiotic prescription in TH and NTH; But the rate of change is higher in NTH.

This change is similar to increase in number of fracture cases. 

However, 

Is there an increase in operations performed?/ or is there any trend? 

Is there an increase in length of hospital stay/ or is there any trend?

Is there any change in mortality/survival?

Authors’ Response: Thank you for these questions. Comparison of trends of antibiotic prescription (more specifically 3rd generation cephalosporins) and trends of fractures was the way to study if antibiotic prescriptions were justifiable or not.

We have now taken your suggestions into account and investigated the trends of performed operations, length of hospital stay and the number of deaths. 

We had total 8 patients (4 in each hospital); who died during 10 years of the study period; therefore, the trend of mortality rate could not be properly assessed. Thus, we assessed here the trend of number of deaths over 10 years as a proxy for mortality rate.

The trends of number of operations, length of hospital stay and number of deaths over 10 years study period at the two study hospitals are presented below; the numbers of rate of change (β) and p-values are given as follows:

 TH NTH

 β p-value β p-value

Operations 0.008 <0.001 0.006 <0.001

Hospital stay -0.013 0.004 -0.04 <0.001

Deaths 0.006 <0.001 -0.00002 0.097

From this calculation it is visible that the number of deaths slightly increased in the TH (β=0.006, p<0.001) and very slightly decreased in the NTH (β= -0.00002, p=0.097) over 10 years; however, this result is not statistically significant.

As per your suggestion regarding the trends of performed operations and length of hospital stay, and due to the results obtained from the calculation above, we have now modified and added the following text:

In methods- page 9, line 147:

“Finally, the analysis of temporal trends in each hospital provided additional justification for

the antibiotic prescribing, by assessing if the percentage of prescribed antibiotics was supported by the percentage of inpatients, and if the percentage of the most frequently prescribed antibiotic class was supported by the percentage of the most common orthopedic diagnosis. Other possible reasons for trends of antibiotic prescription were investigated, such as number of performed operations and length of hospital stay.”

 In results- page 15, line 226: 

In both hospitals, there was a very slight increase in number of performed operations (TH-β=0.008, p<0.001 and NTH-β=0.006, p<0.001) and slight decrease in length of hospital stay (TH-β= -0.013, p=0.004 and NTH-β= -0.04, p<0.001) over 10 years.

In discussion – page 19, line 323: 

General increase in rate of prescription of 3rd generation cephalosporins in both hospitals followed the increase in number of fracture cases over 10 years. Additionally, the increase in prescriptions of 3rd generation cephalosporins could be explained by slight increase in performed operations, although the rates of increase of antibiotic prescriptions in both hospitals were 1.3 and 5 times higher (TH-β=0.01; NTH-β=0.03) than the rates of increase in the performed operations (TH-β=0.008; NTH-β=0.006). Furthermore, the length of hospital stay slightly decreased in both hospitals over 10 years, so this did not support the increase in prescription of 3rd generation cephalosporins.

---

## [Decision Letter · Decision Letter 1]

11 Jan 2021

Trends and patterns of antibiotic prescribing at orthopedic inpatient departments of two private-sector hospitals in Central India: A 10-year observational study

PONE-D-20-33344R1

Dear Dr. Sharma,

We’re pleased to inform you that your manuscript has been judged scientifically suitable for publication and will be formally accepted for publication once it meets all outstanding technical requirements.

Kind regards,

Iddya Karunasagar

Academic Editor

PLOS ONE

Additional Editor Comments (optional):

All reviewer comments have been addressed satisfactorily.

Reviewers' comments:

Reviewer's Responses to Questions

**Comments to the Author**

1. If the authors have adequately addressed your comments raised in a previous round of review and you feel that this manuscript is now acceptable for publication, you may indicate that here to bypass the “Comments to the Author” section, enter your conflict of interest statement in the “Confidential to Editor” section, and submit your "Accept" recommendation.

Reviewer #2: All comments have been addressed

2. Is the manuscript technically sound, and do the data support the conclusions?

Reviewer #2: Yes

3. Has the statistical analysis been performed appropriately and rigorously? 

Reviewer #2: Yes

4. Have the authors made all data underlying the findings in their manuscript fully available?

Reviewer #2: Yes

5. Is the manuscript presented in an intelligible fashion and written in standard English?

Reviewer #2: Yes

6. Review Comments to the Author

Reviewer #2: This study addresses the pattern and trend of antibiotic prescribing over a 10 year periods in a teaching and non teaching hospital in India.

The authors have performed additional analysis as per suggestions.

All comments have been addressed satisfactorily by the authors.

7. PLOS authors have the option to publish the peer review history of their article (what does this mean?). If published, this will include your full peer review and any attached files.

Reviewer #2: **Yes: **Dr Kavitha Saravu

---

## [Editor Report · Acceptance letter]

15 Jan 2021

PONE-D-20-33344R1 

Trends and patterns of antibiotic prescribing at orthopedic inpatient departments of two private-sector hospitals in Central India: A 10-year observational study 

Dear Dr. Sharma:

I'm pleased to inform you that your manuscript has been deemed suitable for publication in PLOS ONE. Congratulations! Your manuscript is now with our production department. 

Kind regards, 

on behalf of

Dr. Iddya Karunasagar 

Academic Editor

PLOS ONE